# Evaluation of Occupational Safety Courses Given in Undergraduate Departments in Turkey and Investigation of the Effect on Working Life

Ismail Ozen [1], Mustafa Yilmaz [2] and Ilyas Kartal [3,*]

1   Occupational Safety Department, Institute of Pure and Applied Sciences, Marmara University, Istanbul 34722, Turkey; ismailozen.isg@gmail.com
2   Mechanical Engineering, Faculty of Engineering, Marmara University, Istanbul 34722, Turkey; mustafa.yilmaz@marmara.edu.tr
3   Metallurgy and Materials Engineering, Faculty of Technology, Marmara University, Istanbul 34722, Turkey
*   Correspondence: ilyaskartal@marmara.edu.tr

**Abstract:** In this study, using the general survey model, which is one of the quantitative research methods, the evaluation of the occupational safety courses given in undergraduate departments in Turkey and their effects on working life were examined for 440 people who received their undergraduate education and started their working life. An appropriate sampling method was preferred in the sample selection of the study. According to the research, the effect of occupational health and safety courses on working life, the contribution of occupational health and safety education, and occupational health and safety education's contribution to awareness show significant differences according to age, gender, department, and work experience. When the relationship between them is examined, these relationships are positive and moderate. The results and survey average scores show that the theoretical courses on occupational health and safety in undergraduate programs of universities in Turkey are insufficient and should be supported by practice and training in working life.

**Keywords:** occupational health and safety; occupational safety courses; working life

## 1. Introduction

Occupational health and safety is a very important phenomenon when occupational accidents and occupational diseases occurring in the world and Turkey are examined. When the research on occupational health and occupational diseases is examined, it is revealed how important material and moral losses, deaths due to accidents and diseases, and permanent incapacitations are in terms of society and public order. Accidents or occupational diseases that occur due to reasons arising from the neglect of occupational safety practices in workplaces not only prevent individuals from continuing their working lives but also create financial difficulties in terms of hospital expenses and compensation to be paid by the workplace. In this case, both workplaces and employees should receive occupational health and safety training and pay attention to their practices. The number of occupational accidents and occupational diseases in countries may vary depending on various factors. The forms of industrialization in countries, awareness in society about occupational health and safety, the level of investigation of accident statistics, and inspection mechanisms are considered factors that affect the number of occupational accidents and occupational diseases [1].

Industry in Turkey is usually concentrated in low-tech classrooms. Occupational groups in the low-tech category are mostly classified as dangerous jobs in terms of occupational health and safety. While low-tech causes increased workplace accidents, high-tech categories aim to minimize job accidents. Accordingly, occupational accidents and diseases are common in such workplaces. It is important to implement training within the scope of

the employer and to implement occupational health and safety controls effectively in order to prevent risks that are outside the acceptable level for workplaces in Turkey [2].

Occupational health and safety training is required to be provided by universities, foundations, professional organizations that are public institutions, educational units of public institutions, and Joint Health and Safety Units (OSGB) authorized by the ministry within the framework of Law No. 6331 in Turkey. The conditions for conducting the tours have been determined by the Ministry of Labor and Social Security. In addition, risks and hazards in the workplace necessitate occupational health and safety [3].

Education undoubtedly has the most important role in the formation of preventive activities and an occupational health and safety culture in Turkey. The majority of occupational accidents are due to a lack of training [4]. For this reason, the "Occupational Health and Safety" course has been compulsory in higher education institutions in Turkey that train people who can be occupational safety experts in the context of the "Occupational Health and Safety Law No. 6331 dated 20 June 2012" by making legal arrangements in paragraph (1) of Article 5 of the Higher Education Law No. 2547 with Law No. 6645 dated 4 April 2015. In this way, the aimed is to create this awareness by offering occupational health and safety courses in faculties [5].

In this study, the importance of OHS was emphasized, occupational health and safety courses given in undergraduate departments in Turkey were evaluated, the training periods and curricula of the institutions were examined, the deficiencies of occupational health and safety training were determined, their effects on working life were discussed, and suggestions were made.

In the research, the aim is to examine the effect of occupational health and safety courses on working life, whether the contribution of occupational health and safety education and occupational health and safety education to awareness differs according to some variables and to determine whether there is a relationship between them.

In this respect, the research will contribute to the literature in terms of examining the effect of occupational health and safety courses on working life in Turkey and presenting suggestions for both educators and students who receive occupational health and safety training as a result of the research in terms of evaluating education within the scope of different variables.

The study was conducted using one of the quantitative research methods, namely, the general survey method. The sample size of the study consisted of 440 people selected from both undergraduate students and people who are in post-graduate working life. The convenience sampling method was used in sample selection. The convenience sampling method is based on items that are completely ready, fast and simple to access. It is the most widely used sample selection method in research. In the study, data were collected with the help of a questionnaire form and analyzed with the SPSS 22.0 program.

## 2. Occupational Health and Safety

It brings many problems to working life. Perhaps the most important of these problems is unhealthy and unsafe working conditions. If precautions are not taken, it is an inevitable fact that these conditions cause occupational accidents and diseases. Taking measures and precautions to protect employees from accidents and diseases is addressed within the scope of occupational health and safety [6].

The protection of workers is a fundamental issue in labor law. However, the main situation for the protection of workers is to take precautions before occupational accidents and diseases occur. From this point of view, preventive studies should cover all processes and consider the environments in which workers work as a whole [7].

The World Health Organization (WHO) and the International Labour Organization (ILO) have defined the concept of occupational health as the full well-being of an employee, not only physically but also mentally and socially, and ensuring the sustainability of his or her situation in the best conditions. In other words, occupational health is aimed at

workers being free from negative situations and conditions in the environment in which they work and to ensure that the work and the worker are in harmony [8].

While the concept of safety refers to being safe alone, the concept of occupational health and safety means being away from damage and unacceptable risks. Staying away from irreparable damages and risks is the basis of the concept of occupational health and safety. Uncompensated damages and risks can cause occupational diseases and occupational accidents in employees. For this reason, every business or company should have its own occupational health and safety culture. Occupational health and safety, which is expressed as the measures carried out or obliged to be carried out in order to minimize or prevent the risks and hazards that may occur in the working environment, means that these situations are eliminated in order to prevent employees from facing any negative situations. Most employees worldwide spend at least 1/3 of their lives in the workplace. Through tax revenues, the material basis and economy of society are maintained. However, thousands of working hours are lost every year due to occupational accidents. According to global estimates, occupational diseases and accidents take more than 2 million lives per year, while approximately 160 million diseases and 337 million accidents occur per year [9].

Safety and health conditions are very important for employees. OHS is important for employees as well as employers and society. The lives and futures of employees may sometimes be threatened due to work accidents and occupational diseases. Working individuals need to avoid the dangers and risks that increase with industrialization, especially the dangers to their lives, health, and body integrity. Globally, the majority of employees spend a large part of their lives in the workplace. However, thousands of working hours are lost worldwide every year due to work accidents and occupational diseases. Occupational accidents, occupational diseases, and injuries have a huge economic burden on society. When the statistics between 2013 and 2021 in Turkey are examined, it is observed that there is an increase in occupational accidents every year except 2020 [10].

Employees in businesses that lack occupational health and safety practices show a low performance in terms of productivity. Disruptions in production as a result of occupational accidents or diseases and compensations to be paid to workers will have a much higher cost than occupational health and safety practices. As a result of the measures taken, the gains to be made by the employers will be much less costly than the repercussions to be encountered if the employee has an occupational accident. The training provided on this subject is extremely effective. Within the scope of an article examining the employee activities before and after the training given to the employees and published in the United States in 2004, the effectiveness of the training given to the employees with the help of different surveys was discussed. In the study, it was revealed that the activities of employees to make changes after the training increased significantly compared to the time before the training and they were more willing to change the field conditions. In addition, this study reveals that it increases the willingness of workers to make improvements in issues such as safety and health and their self-confidence in making these improvements [11].

When the studies in the literature are examined, it is seen that there are studies examining occupational health and safety education within the scope of different variables. While some of these studies show similarities with the study we discussed, others do not. It is important to examine the studies in the literature and make the missing points the subject of new studies.

Alkan (2017) conducted a survey of 693 students in his study. As a result of the analysis of the information obtained, it was seen that the knowledge levels of the culture related to occupational health and safety differed. It has been determined that the difference between the awareness and culture levels of the students regarding occupational health and safety compared to their education level is not statistically significant [12].

Karal (2018), in his study, applied a questionnaire to 438 students. As a result of this study, factor analysis, Barlett's test, item analysis, item-total score correlation, etc., analysis studies were applied. As a result of the analysis studies, it was determined that the students'

awareness of occupational health and safety was at a high level and there was a positive relationship in the sub-dimensions of the awareness scale [13].

A descriptive field study was conducted by Yenisarı et al. (2019) to determine the awareness levels of employees regarding OHS training. In the study, literature review and survey methods, which are quantitative research techniques, were used. The survey was conducted using face-to-face interviews with academic and administrative staff working in different units of Çanakkale Onsekiz Mart University. The answers given by the participants to the questionnaires were subjected to confidence analysis using the Cronbach's alpha test. As a result of the research, approximately 12.8% of the participants stated that the causes of occupational accidents were due to lack of education [14].

In the study by Küçükoğlu (2018), a survey was conducted with 156 students to determine their awareness of occupational health and safety. It was concluded that the students perceived the objectives, results, and scope of occupational health and safety positively, but they did not have a good grasp of the subjects related to occupational health and safety. It was determined that the students could not perceive occupational health and safety as a science branch [15].

In Pınar's (2018) study, the aimed was to determine the contribution of education to the formation of occupational health and safety. In line with the aforementioned purpose, a survey was conducted with students in a vocational high school within the scope of the school project. As a result of the analysis, it was determined that the culture of occupational health and safety was not formed in the students [16].

Systematic and multidimensional studies are needed to improve efficiency in the field of occupational health and safety in Turkey. It is essential that the audit and coordination studies in the sectors be examined. In this context, it is revealed that additional institutional arrangements are needed in the studies conducted. Accordingly, system analysis and design methods should be applied to investigate the problems. In addition, solutions need to be proposed to improve the existing institutional working standards system [17].

## 3. Training in Occupational Health and Safety

It is a natural right for people to work in a healthy and safe environment. The social law state is responsible for protecting and securing the rights of its citizens. Numerous factors can be listed in order to create a safe and healthy environment. However, training is a more important issue than other factors in terms of employees' learning about risks and hazards. Training is very important in terms of eliminating all problems that may occur in the working environment. As a matter of fact, there is a need for training on occupational health and safety in order for employees to understand and avoid the hazards and risks they may encounter in their workplaces. However, the fact that occupational health and safety training is provided only due to legal requirements and does not become a workplace culture will cause this training to be ineffective, thus paving the way for the loss of time and labor [18].

The main purpose of occupational health and safety training is to create positive behavior change. However, having knowledge about a subject does not mean following the best practices in that regard. For example, knowing that smoking is harmful to health and still continuing to smoke is a good example of this situation. In this context, not only education but also its effectiveness are considered important issues in terms of occupational health and safety [19].

The effectiveness of occupational health and safety concerns not only the worker segment but also the manager and employer segments. The more effective occupational health and safety training is, the more conscious behaviors will increase, productivity will increase, and there will be a decrease in occupational diseases and accidents. Therefore, the opinions of employers and managers on occupational health and safety education are qualified to direct the behaviors of employees. Providing effective, accurate, and real training will have a decisive effect on the perceptions of employees. On the other hand, arousing the need for education, explaining that the information given is important in

terms of safety and health in daily life, enriching the subjects with visuals, and arousing the interest of the learner are seen as the responsibilities of employers and managers [20].

Another subject of education is making learning fun, regardless of age group. Therefore, the training to be given should be well planned. In education, time should be used efficiently, given in the form of controllable groups, and turned into fun. Supporting education with visuals and increasing participation in the form of questions and answers, as well as presenting sections from experiences and life, can be considered factors that will facilitate learning. Another point about education is that legal rights and responsibilities are taught to individuals. The knowledge of the employees on this subject will increase their interest in occupational health and safety training. Within the scope of all of these, it is revealed that occupational health and safety training will increase the interest of employees in safety and health, increase their level of interest, show positive behavioral changes, and decrease occupational disease and occupational accidents [21].

The role of occupational health and safety training in the prevention of occupational accidents and diseases is a subject that has been included in the literature. Although legal regulations are made regarding occupational accidents and diseases, the consequences of mistakes made by employees can be very severe. The most effective way to eliminate dangerous and risky behaviors by employees is to provide training. Occupational health and safety training comes into play when employees are unskilled or uninformed. On the other hand, training aimed at establishing self-control mechanisms in personnel is the best and most cost-effective way to prevent occupational accidents and diseases [22].

Habits are the basis of human behavior. At the same time, this is another important issue that is present in working life. People who have been brought up without occupational health and safety training have the perception that the occupational accidents and diseases they encounter are a necessity of the job they do. Employees who encounter or witness such incidents tend to act more courageously, having the idea that nothing will happen to them over time. As a matter of fact, the vast majority of occupational accidents and occupational diseases occur due to this mindset [23].

OHS training enables employees to work as owners of willpower. These trainings, which are extremely important for the creation of self-control mechanisms in employees, are seen as the biggest factor in the formation of responsible behaviors requested from employees. They aim to provide people with competencies such as socialization, self-acceptance, responsibility, and respect. Within the scope of the trainings on occupational health and safety offered to people, when people feel that their own health will be protected directly, their participation may also increase within the scope of their appreciation of this situation. In addition, it is possible to mention that employees participate in the learning process, and their enthusiasm for learning increases when the studies for human health and welfare are supported [24].

In addition to all these, occupational health and safety education is also very important in terms of the sustainability of internal power. As a result of the accidents faced by people working in risky professions while carrying out a job, things may be disrupted. Therefore, it can be said that receiving occupational safety training directly affects workforce sustainability.

### 3.1. Occupational Health and Safety Training in Undergraduate Departments in Our Country

Creating a culture within the scope of OHS will contribute to increasing the safety awareness of workplace employees. Thanks to OHS training, employee behavior can be controlled, and the employee can be provided with the status of not leaving the herd. In this way, safe behaviors will be maximized and unsafe behaviors will be minimized. Workplaces should participate in OHS trainings to raise OHS awareness [25].

Occupational health-related courses have been included in the curriculum of medical schools since the 1960s. Initially, the courses given between 9 and 15 h in these faculties started to be given as 2–3 h in the 1st, 3rd, 5th, and 6th grades of 6-year medical faculties, and then in architecture, science, and some other departments of state universities and

foundation universities. Pursuant to Article 69 of Bag Law No. 6645 dated 2015, this course has been compulsory for vocational high schools and faculties of universities that provide graduates in a situation that may be occupational health and safety [26].

OHS training is given in the colleges and faculties of universities that provide 4-year education. At the same time, 4-year OHS training is provided in the open education facility. Occupational health and safety training in Turkey was first started within Yeni Yüzyıl University [27]. Graduates of engineering, science, and architecture faculties and technical teachers, as well as graduates of occupational health and safety undergraduate or associate degree programs, can become occupational safety specialists by taking the OSYM exam. Engineering faculty graduates are entitled to take the exam after receiving 220 h of training and 40 h of internship at OHS training institutions authorized by the Ministry of Labor and Social Security. Candidates who reach 70 points in the exam are entitled to receive the occupational safety specialist certificate (Occupational Health And Safety Law) [5,26].

When the curriculum of Yeni Yüzyıl University is examined, the following can be noted: "introduction to occupational health and safety, medical first aid, health and safety culture, physical, chemical, biological, and psychosocial risk factors, occupational health and safety legislation, materials and strength, occupational physiology, ergonomics, fire safety, toxic and dangerous industrial substances, occupational hygiene and laboratory techniques, occupational health and safety management systems, risk analysis and evaluation, periodic controls and maintenance and repair, occupational health and safety in mining works, and dangerous goods transportation."

Students who have completed four years of education can enter the C-class occupational safety specialty exam without attending any courses, unlike those who have an associate degree. Those who successfully complete the exam receive the title of occupational health and safety specialist. Occupational health and safety education is provided at the undergraduate level at the university, which is located in many provinces and districts in Turkey. As of 2020–2021, the number of students receiving OHS education at the undergraduate level in Turkey was 13,591 [28].

When the curricula of the universities within the scope of occupational health and safety are examined, it is seen that a standard application cannot be achieved. The reason for this situation is that the awareness of occupational health and safety culture in Turkey and universities is not yet established, and there are insufficient lecturers. The occupational health and safety training curriculum prepared for higher education graduates to obtain an occupational health and safety expert certificate in Turkey was organized by the "Ministry of Labor and Social Security" [29].

The occupational health and safety education curriculum determined by the Ministry of Labor and Social Security has been organized as a result of research conducted in many countries. In this context, it is determined that occupational health and safety courses taken for at least two semesters in faculties such as engineering cannot be transferred within this period. Occupational health and safety courses should be extended to at least four semesters in order to effectively provide all the courses in the curriculum by the Ministry of Labor and Social Security.

### 3.2. The Effect of Occupational Health and Safety Training on Working Life

Informing employees and providing training is extremely important in terms of OHS. Focusing on OHS issues also has an important place in terms of showing the value given to employees. In order to maintain occupational health and safety, managers and employers are obliged to inform their employees about security risks, extraordinary situations, preventive and protective measures, fires, and natural disasters that they may encounter in the workplace [30].

In the Safety Culture Report published by the International Labour Organization in 2012, 80% of occupational accidents and all occupational diseases are preventable. In addition, the great source of the problems of the unconscious and contrary occupational safety behaviors of the employees has been evaluated to be a lack of education [31].

When the occupational accidents occurred, it was concluded that the vocational training given to the workers was not sufficient, the necessary infrastructures were not established, there were generally low-level employees in risky sectors, and in-service training and on-the-job training were not sufficient. In this context, it has been revealed that education is of great importance in preventing occupational accidents. In order to overcome the current problem, it has been made mandatory to provide employers with occupational health and safety training for employees. It is emphasized that the training provided in the field of OHS should be given especially before the employee starts work, in case of a change of work equipment, and in case of a change of work and workplace. The training given should be renewed and continued periodically against the risks that change and occur depending on technological developments [32].

Employers are also obliged to fulfill their duty to obtain the opinions of their employees and to participate. Employers should take into account the opinions of the employees on occupational health and safety, ensure their participation, apply new technology tools to working conditions, evaluate the preference of the work equipment to be selected, and evaluate the effect of the working environment conditions on occupational health and safety by taking the opinions of the employees into account. Especially in the case of risk assessment, taking into account the opinions of the employees, reporting the risks in practice, and eliminating the desired risks have a great effect on preventing the occurrence of occupational accidents. A safe working environment can also help reduce business interruptions and costs and strengthen the image of the organization [33].

Providing OHS training in workplaces is extremely important in terms of providing safe behaviors to employees. In addition, effective and correct methods should be applied in the training provided. As the most effective method in the development of behavioral skills, the experiential learning method, that is, employees see the problems they will encounter at work in practice and record them in long-term memory. Knowledge alone is not enough to change behaviors. Information needs to be perceived, experienced, and stored in long-term memory in order to transform into behavior in the event of danger. As a result of all these, it can be mentioned that the effects of occupational health and safety training on working life are undeniably high. Education is among the most important components of occupational health and safety. It is stated that there is evidence showing that companies with "Occupational Health and Safety" performance, which includes effective training, have the characteristic that education is a positive safety culture [34].

It should be known that the creation of training opportunities is not only necessary to fulfill the basic rights of employees to be protected from risks in the workplace but also a legal necessity for the judiciary and should be acted upon accordingly [35].

Occupational health programs and common sense are elements that ensure the complete elimination of hazards as the best and most effective form of solution in order to eliminate the exposures arising from the dangerous work environment. Accordingly, priority is an issue that needs to be recognized. When the risks are not eliminated, controlling the workers at the highest possible level in order not to be exposed to the risks that cannot be eliminated is considered an extremely desirable approach. The training received by the employees is not a means of control in order to eliminate the risks [36].

In the study conducted by Yildirim et al. (2022), it was stated that there are problems in terms of working standards and application, especially in the construction industry in Turkey. Within the scope of the study, the institutional problems of the current system were investigated, and cause-effect analysis and stakeholder analysis were applied. Within the scope of the reports and findings collected from the literature and experts within the scope of the study, it has been revealed that there is still no maturity in issues such as occupational safety, measures against occupational accidents and diseases, working time, inspection, and control in Turkey. In this regard, it has been determined that systematic perspectives are needed to develop holistic, multidimensional, audit, and coordination studies in many sectors [17].

In the study conducted by Laberge et al. (2014), especially the occupational accidents frequently experienced by young employees and the training strategies created to prevent these accidents are generally considered ineffective. In this context, occupational health and training within the organization should consist of the following stages [36]:

- Determination of education policies,
- Selection of training techniques,
- Designing training plans,
- Making determinations about the goals and objectives of the training,
- Identifying and defining the basic needs of education.

Determining who needs to be trained will improve performance in the field of occupational health and safety and prevent accidents and occupational diseases. In this sense, it would be appropriate to mention that the effect of occupational health and safety training on business life is undeniably high [37].

In the study, the aimed was to examine whether the effect of occupational health and safety courses on working life, occupational health and safety education, and the contribution of occupational health and safety education to awareness differ according to some variables and to determine whether there is a relationship between them. The hypotheses developed for this are as follows:

**H1.** *There is a statistically significant difference between the effect of occupational health and safety courses on working life, occupational health and safety training, and the contribution of occupational health and safety training to awareness according to age.*

**H2.** *There is a statistically significant difference between the effect of occupational health and safety courses on working life, occupational health and safety education, and the contribution of occupational health and safety education to awareness according to gender.*

**H3.** *There is a statistically significant difference between the effect of occupational health and safety courses on working life, occupational health and safety education, and the contribution of occupational health and safety education to awareness according to department.*

**H4.** *There is a statistically significant difference between the effect of occupational health and safety courses on working life, occupational health and safety education, and the contribution of occupational health and safety education to awareness according to work experience.*

**H5.** *There is a statistically significant relationship between the effect of occupational health and safety courses on working life, occupational health and safety training, and the contribution of occupational health and safety training to awareness.*

## 4. Method

In the study, the quantitative research method was used in the evaluation of occupational safety courses offered in undergraduate departments in Turkey. Their effects on working life were examined on a designed scale. The study also aimed to examine whether the effect of occupational health and safety courses on working life, occupational health and safety education, and the contribution of occupational health and safety education to awareness differ according to some variables and to determine whether there is a relationship between them.

### 4.1. Research Model

While conducting scientific research, various research methods were used to organize the study, determine the data collection tools, obtain data, create findings and base the results obtained as a result of the research on a scientific basis. Quantitative research is a type of research that reveals facts and events in an observable, measurable, and numerically meaningful form by removing them from subjectivity and objectifying them. In the quantitative research model, numerical results are obtained from the sample that will represent the research population. In these studies, the orientation of the research

population's views on the research subject is examined [38]. At this point, the part to be considered is that the sample that will represent the research population should be selected correctly and the correct questions should be directed to this selected sample. In this context, it was found appropriate to use the survey method, which is one of the quantitative research methods, in this study.

In addition, in quantitative studies, determining the demographic characteristics of the sample taken from the research population, their views on the determined subjects, and their preferences on the subject is attempted [39]. In this respect, the difference between the total scores according to the variables determined in the study was examined.

*4.2. Population and Sample*

The sample size of the study consists of both undergraduate students and people who are in postgraduate working life. The sample size of the study consisted of 440 people with an average age of $27.83 \pm 8.992$ years, selected from both undergraduate students and post-graduate students. In the sample selection, the convenient sampling method was used. The convenient sampling method relies entirely on ready, fast, and easy-to-access items. It is the most commonly used sample selection method in research [40]. It is based on ready-made, fast, and easy-to-access items since random selection is made from a crowded universe and the most easily accessible participants constitute the sample. Since it was determined in our study that individuals with occupational health and safety training formed a crowded universe, making a random selection among them shows that convenient sampling is the best sampling strategy for this study. In the research, it is taken into account where the most information can be obtained from the limited number of cases to be sampled, and the sample is selected accordingly [41]. In cases where the population is more than 500 thousand people, it is stated that a sample group of 300 people is sufficient to represent the universe [42]. Therefore, conducting the study with 440 participants was deemed sufficient to represent the population. In addition, it was determined that the sample of the study was sufficient to represent the population with the G*Power analysis performed with 95% confidence and 5% margin of error.

*4.3. Data Collection Process and Tools*

The survey method was used as a data collection tool to determine the participants' perceptions about the evaluation of occupational safety courses offered in undergraduate departments in Turkey and their impact on working life. The questionnaire form consists of 4 sections. The first section includes 9 questions about the demographic information of the participants, the second section includes 14 survey questions about the impact of occupational health and safety courses on working life, the third section includes 15 survey questions about occupational health and safety education, and the fourth section includes 17 survey questions about the contribution of occupational health and safety education to awareness. The questions used were scored on a 5-point Likert. These questionnaires used do not consist of sub-dimensions but consist of a single dimension. The necessary ethics committee and research approval was obtained for the study, and the participants were included in the study after they stated that they volunteered to participate in the study. Participants evaluated the survey form via Google Forms.

*4.4. Data Analysis*

One of the important processes to be considered while conducting scientific research will be the analysis of data. The data obtained in the research should be analyzed and interpreted correctly with data analysis methods suitable for the research and in line with the purposes of the research. Depending on whether the data obtained from the research are quantitative or qualitative, the researchers should have information about some important details of the process in question. Therefore, SPSS was used in the data analysis of the study. As a result of the analyses, the Cronbach's alpha coefficients of the impact of occupational health and safety courses on working life, occupational health and

safety training, and contribution of occupational health and safety training to awareness were 0.71, 0.88, and 0.88, respectively. According to Cronbach's alpha coefficient, a scale should have a Cronbach's alpha value of 0.70 and above to be considered reliable [43]. The obtained coefficient values greater than 0.70 indicate that the scales used in this study are highly reliable. The kurtosis and skewness values obtained from the scale scores between +3 and −3 are considered sufficient for normal distribution. In addition, histograms, graphs, and coefficients of variation for the relevant scale and its dimensions were also examined and found to be suitable for normal distribution. In this context, parametric methods were used in the analyses. The ANOVA test was used in the analyses according to age and department variables, independent samples *t*-test was used in the analyses according to gender and work experience status, and Pearson correlation test was used in their relations with each other.

## 5. Results

In this part of the study, the data obtained from the research are discussed and the results interpreted.

Of the participants, 216 were students, 160 were graduates, 42 were academicians, and 22 were employers. Furthermore, 222 were between the ages of 17–25, 139 were between the ages of 26–35, and 79 were 36 and over. Moreover, 333 of the participants were male and 107 were female. Of the 440 participants, 52 of them were from the departments of science, 70 from the departments of occupational health and safety, 21 from the departments of architecture, and 297 from the departments of engineering. Additionally, 300 had work experience and 140 had no work experience. While 60 of them had experienced an occupational accident in their working life, 380 of them had not. While 42 of them had experienced an occupational accident in their education life, 398 of them had not. While 138 people had experienced near misses during training, 302 people had not experienced near misses. While 225 people answered the biggest cause of occupational accidents as unsafe (unsafe) behavior of the employee, 215 people answered as unsafe (unsafe) work environment (Table 1).

**Table 1.** Findings related to the demographic information of the participants.

|  |  | Frequency (*n*) | Percentage (%) |
|---|---|---|---|
| Occupational Status | Student | 216 | 49.1 |
|  | Graduate | 160 | 36.4 |
|  | Academic | 42 | 9.5 |
|  | Employer | 22 | 5.0 |
| Age | 17–25 years old | 222 | 50.5 |
|  | 26–35 years old | 139 | 31.5 |
|  | Age 36 and over | 79 | 18.0 |
| Gender | Male | 333 | 75.7 |
|  | Female | 107 | 24.3 |
| Department of Service | Science | 52 | 11.8 |
|  | Occupational Health and Safety | 70 | 15.9 |
|  | Architecture | 21 | 4.8 |
|  | Engineering | 297 | 67.5 |
| Work Experience Status | I have work experience | 300 | 68.2 |
|  | I have no work experience | 140 | 31.8 |

**Table 1.** *Cont.*

|  |  | Frequency (*n*) | Percentage (%) |
|---|---|---|---|
| Occupational Accident at Work | Yes | 60 | 13.6 |
|  | No | 380 | 86.4 |
| Occupational Accident During Training | Yes | 42 | 9.5 |
|  | No | 398 | 90.5 |
| Near Miss Event Experiencing During Training | Yes | 138 | 31.4 |
|  | No | 302 | 68.6 |
| The Biggest Cause of Occupational Accidents | Unsafe (unsafe) behavior of the employee | 225 | 51.1 |
|  | Work environment without occupational safety (unsafe) | 215 | 48.9 |
| Total |  | 440 | 100 |

Graduates include individuals who have graduated from engineering, architecture, science, and occupational health and safety departments without work experience and individuals who have previous work experience through internships, etc. who are not currently actively working.

According to the one-way variance test conducted to compare the participants according to their ages, statistically significant differences were found between the effect of occupational health and safety courses on working life and occupational health and safety training ($p < 0.05$). While the effect of occupational health and safety courses on working life and the scores of occupational health and safety education of the participants vary according to their ages, the scores of the contribution of occupational health and safety education to awareness do not change according to their ages. Accordingly, the effect of occupational health and safety courses on working life of those aged 36 and over is higher, and the total average of occupational health and safety education of those aged 26–35 is higher (Table 2).

**Table 2.** Findings regarding the comparison of participants by age.

|  |  | *n* | Average | Std. Deviation | F | *p* |
|---|---|---|---|---|---|---|
| The Effect of Occupational Health and Safety Courses on Working Life | 17–25 years old | 222 | 47.4324 | 7.48346 | 5.827 | 0.003 |
|  | 26–35 years old | 139 | 48.6115 | 4.91155 |  |  |
|  | Age 36 and over | 79 | 50.2152 | 4.97602 |  |  |
|  | Total | 440 | 48.3045 | 6.42234 |  |  |
| Occupational Health and Safety | 17–25 years old | 222 | 53.1036 | 10.05114 | 17.443 | 0.000 |
|  | 26–35 years old | 139 | 57.9137 | 6.30445 |  |  |
|  | Age 36 and over | 79 | 57.8101 | 6.77302 |  |  |
|  | Total | 440 | 55.4682 | 8.78697 |  |  |
| Contribution of Occupational Health and Safety Training to Awareness | 17–25 years old | 222 | 57.1892 | 10.62611 | 0.508 | 0.602 |
|  | 26–35 years old | 139 | 56.8417 | 8.92908 |  |  |
|  | Age 36 and over | 79 | 58.2025 | 8.12937 |  |  |
|  | Total | 440 | 57.2614 | 9.68842 |  |  |

According to the independent groups *t*-test conducted to compare the participants according to their gender, statistically significant differences were found between them and occupational health and safety training ($p < 0.05$). While the occupational health and safety education scores of the participants vary according to their genders, the effect of

occupational health and safety courses on working life and the contribution of occupational health and safety education to awareness scores do not change according to their genders. Accordingly, the total average of occupational health and safety education of women is higher (Table 3).

**Table 3.** Findings regarding the comparison of the participants by gender.

|  | Gender | *n* | Average | Std. Deviation | t | *p* |
|---|---|---|---|---|---|---|
| The Effect of Occupational Health and Safety Courses on Working Life | Male | 333 | 48.1261 | 6.62841 | −1.108 | 0.269 |
|  | Female | 107 | 48.8598 | 5.72721 |  |  |
| Occupational Health and Safety | Male | 333 | 54.6757 | 9.08444 | −3.776 | 0.000 |
|  | Female | 107 | 57.9346 | 7.29211 |  |  |
| Contribution of Occupational Health and Safety Training to Awareness | Male | 333 | 56.8589 | 9.90989 | 1.540 | 0.124 |
|  | Female | 107 | 58.5140 | 8.89176 |  |  |

According to the one-way variance test conducted to compare the participants according to their departments, statistically significant differences were found between the effect of occupational health and safety courses on working life, occupational health and safety training, and the contribution of occupational health and safety training to awareness ($p < 0.05$). The effect of occupational health and safety courses on working life, occupational health and safety education, and the contribution of occupational health and safety education to awareness scores of the participants vary according to their departments. Accordingly, the total averages of the effects of occupational health and safety courses on working life, occupational health and safety education, and occupational health and safety education on awareness are higher in those who are in the occupational health and safety department (Table 4).

**Table 4.** Findings regarding the comparison of participants by departments.

|  |  | *n* | Average | Std. Deviation | F | *p* |
|---|---|---|---|---|---|---|
| The Effect of Occupational Health and Safety Courses on Working Life | Science | 52 | 49.9038 | 5.79134 | 4.622 | 0.003 |
|  | Occupational Health and Safety | 70 | 50.2714 | 7.67061 |  |  |
|  | Architecture | 21 | 47.8095 | 5.97176 |  |  |
|  | Engineering | 297 | 47.5960 | 6.11420 |  |  |
|  | Total | 440 | 48.3045 | 6.42234 |  |  |
| Occupational Health and Safety | Science | 52 | 55.5385 | 8.90354 | 6.985 | 0.000 |
|  | Occupational Health and Safety | 70 | 59.6143 | 8.98677 |  |  |
|  | Architecture | 21 | 52.6667 | 8.35065 |  |  |
|  | Engineering | 297 | 54.6768 | 8.48906 |  |  |
|  | Total | 440 | 55.4682 | 8.78697 |  |  |
| Contribution of Occupational Health and Safety Training to Awareness | Science | 52 | 57.3654 | 8.91799 | 5.319 | 0.001 |
|  | Occupational Health and Safety | 70 | 61.2714 | 10.56371 |  |  |
|  | Architecture | 21 | 58.1429 | 8.44562 |  |  |
|  | Engineering | 297 | 56.2357 | 9.47427 |  |  |
|  | Total | 440 | 57.2614 | 9.68842 |  |  |

According to the independent groups *t*-test conducted to compare the participants' work experience, statistically significant differences were found between the effects of occupational health and safety courses on working life and occupational health and safety

training ($p < 0.05$). While the impact of occupational health and safety courses on working life and occupational health and safety training scores of the participants vary according to their work experience status, the contribution of occupational health and safety training to awareness scores do not change according to their work experience status. Accordingly, the effect of occupational health and safety courses on working life and the total average of occupational health and safety education of those with work experience are higher (Table 5).

**Table 5.** Findings regarding the comparison of participants according to work experience status.

| | Work Experience Status | *n* | Average | Std. Deviation | t | *p* |
|---|---|---|---|---|---|---|
| The Effect of Occupational Health and Safety Courses on Working Life | I have work experience | 300 | 48.8933 | 6.10405 | 2.838 | 0.005 |
| | I have no work experience | 140 | 47.0429 | 6.91144 | | |
| Occupational Health and Safety | I have work experience | 300 | 56.9133 | 7.86971 | 4.800 | 0.000 |
| | I have no work experience | 140 | 52.3714 | 9.82168 | | |
| Contribution of Occupational Health and Safety Training to Awareness | I have work experience | 300 | 57.3567 | 9.07472 | 0.302 | 0.763 |
| | I have no work experience | 140 | 57.0571 | 10.92010 | | |

According to the Pearson correlation test conducted to examine the relationship between the effect of occupational health and safety courses on working life, occupational health and safety education, and the contribution of occupational health and safety education to awareness, it was determined that there was a statistically significant relationship between them ($p < 0.01$). While the effect of occupational health and safety courses on working life affects the contribution of occupational health and safety training and occupational health and safety training to awareness, occupational health and safety training also affects the contribution of occupational health and safety training to awareness. Accordingly, these relations are positive and moderate (Table 6). Therefore, it can be said that as the effect of occupational health and safety courses on working life increases, the contribution of occupational health and safety education and occupational health and safety education to awareness increases, decreases in cases where it decreases, and the contribution of occupational health and safety education to awareness increases and decreases when occupational health and safety education increases.

**Table 6.** Findings regarding the correlation between the effect of occupational health and safety courses on working life, occupational health and safety training, and the contribution of occupational health and safety training to awareness.

| | | The Effect of Occupational Health and Safety Courses on Working Life | Occupational Health and Safety | Contribution of Occupational Health and Safety Training to Awareness |
|---|---|---|---|---|
| The Effect of Occupational Health and Safety Courses on Working Life | r | 1 | 0.626 ** | 486 |
| | *p* | | 0.000 | 0.000 |
| | *n* | 440 | 440 | 440 |
| Occupational Health and Safety | r | 0.626 ** | 1 | 0.401 ** |
| | *p* | 0.000 | | 0.000 |
| | *n* | 440 | 440 | 440 |
| Contribution of Occupational Health and Safety Training to Awareness | r | 486 | 0.401 ** | 1 |
| | *p* | 0.000 | 0.000 | |
| | *n* | 440 | 440 | 440 |

** The correlation is significant at the level of 0.01 (2-tailed).

## 6. Discussion

Within the scope of the study, the findings and results obtained from the data in this part of the study conducted to evaluate the occupational safety courses given in undergraduate departments in Turkey and to examine their effect on working life are discussed.

Statistically significant differences were found between the effects of occupational health and safety courses on working life and occupational health and safety education according to the age of the participants. Accordingly, the effect of occupational health and safety courses on working life of those aged 36 and over is higher, and the total average of occupational health and safety education of those aged 26–35 is higher. Since the knowledge of those between the ages of 26 and 35 about occupational health and safety is more up-to-date since they are at the beginning of their working lives, we think that the average effect of occupational health and safety courses on working life is higher because they are more experienced in occupational health and safety education, and because they are 36 years old and over.

Similar to our study, Şaşar (2022), who examined the effect of occupational health and safety on working life according to age, found a positive and significant relationship between the age of the participants and their views on occupational health and safety [44]. Tombulca (2022) found that the attitudes of radiology paramedics towards occupational health and safety showed a statistically significant difference according to their age [45]. He stated that the attitudes towards occupational health and safety of paramedics whose ages were higher than the others were higher.

Statistically significant differences were found between the occupational health and safety training according to the gender of the participants. Accordingly, the total average of occupational health and safety education of women is higher. We think this is due to the fact that women are generally more sensitive, careful, and meticulous than men.

Unlike our study, Tombulca (2022) determined as a result of his study that the attitudes of healthcare professionals working in radiology towards occupational health and safety did not show a statistically significant difference according to their gender [45]. Aygün (2017) stated that the participants' perspectives on occupational health and safety according to their gender showed statistically significant differences and that the perspectives of female participants regarding occupational health and safety were more positive than male participants [46].

According to the departments of the participants, statistically significant differences were found between the effect of occupational health and safety courses on working life, occupational health and safety training, and the contribution of occupational health and safety training to awareness. Accordingly, the total averages of the effects of occupational health and safety courses on working life, occupational health and safety education, and occupational health and safety education on awareness are higher in those who are in the occupational health and safety department. It is thought that such a result has emerged because those who study occupational health and safety are naturally more knowledgeable and experienced in this regard.

Similarly, Aygün (2017) stated that the participants' perspectives on occupational health and safety according to their departments showed statistically significant differences and that the participants in the occupational health and safety department had a statistically more positive perspective on occupational health and safety compared to other partici-pants [46]. Güngör (2017) stated as a result of his study that the participants' views on occupational health and safety did not show statistically significant differences according to their departments, and that the views of participants from different departments on occupational health and safety were at similar levels [47]. This contradictory situation that arises according to the departments is due to the occupational health and safety department. As in our study, there is a significant difference in the sample, while there are those directly from the occupational health and safety department.

Statistically significant differences were found between the effect of occupational health and safety courses on working life and occupational health and safety training according to the participants' work experience. Accordingly, the total averages of the effect of occupational health and safety courses on working life and occupational health and safety training of those with work experience are higher. It is thought that such a result emerged because those with work experience apply occupational health and safety not only in theory but also in practice.

Similarly, Tombulca (2022) found that the attitudes of radiology paramedics towards occupational health and safety showed a statistically significant difference according to their professional experience [45]. He stated that the attitudes towards occupational health and safety of paramedics with more professional experience than others were higher. Unlike our study, Şaşar (2022) found that there was no significant positive relationship between the participants' working time in their profession and their opinions on occupational health and safety, and that the opinions of people with different professional durations on occupational health and safety were at similar levels [44].

It has been determined that there is a statistically significant relationship between the effect of occupational health and safety courses on working life, occupational health and safety training, and the contribution of occupational health and safety training to awareness. Accordingly, this relationship is positive and moderate. Since these three scales are interconnected, united in the common point of occupational health and safety, and try to determine the knowledge and experiences on this subject, it is thought that such a result has emerged.

Similarly, when Karal (2018) examined the contribution of occupational health and safety education and occupational health and safety education to awareness as a result of her study, it stated that there was a statistically significant relationship between them and this correlation was moderately positive [13]. As a result of his study, Yenisarı et al. (2019) stated that when he examined the relationship between the effect of occupational health and safety courses on working life and occupational health and safety education, there was a statistically significant relationship between them and this correlation was moderately positive [14].

The limitations of the research are: The fact that only participants from Turkey participated constitutes a limitation in terms of generalizability. The use of the quantitative research method in the research and the possibility of accessing more information with the qualitative interview technique constitute a limitation in terms of data collection tools. The fact that a comparison was made in the research according to age, gender, department, and work experience creates a limitation in terms of the variables used in the research. The fact that the research was conducted only with individuals who received occupational health and safety training also creates a limitation in terms of the sample.

## 7. Conclusions and Recommendations

In this study, which aims to evaluate the occupational safety courses given in undergraduate departments in Turkey and to examine their impact on working life, whether taking occupational safety courses within the scope of undergraduate education has an impact on the working lives of people who finish their education and start working life was investigated. In the study, the effects of occupational health and safety courses on working life and occupational health and safety education were examined according to the occupational status, age, gender, and department of the employees. Students from various disciplines with and without work accident experience and people with work experience participated in the survey. Data were collected by both men and women. According to the research, the effect of occupational health and safety courses on working life, the contribution of occupational health and safety education, and occupational health and safety education's contribution to awareness show significant differences according to age, gender, department, and work experience. When the relationship between them is examined, these relationships are positive and moderate. The results obtained and the average scores of the

questionnaire show that the theoretical courses on the protection of health and safety in the workplace in undergraduate programs in universities in Turkey are insufficient and should be supported by practice and training in working life. The fact that the research was conducted only in Turkey and that the data was collected online poses a limitation in terms of generalizability.

Recommendations for the conclusions obtained from the study are as follows:

- It should be ensured that OHS is accepted as a culture by society and for this reason, it is necessary to teach this education effectively to spread it to all education levels starting from primary education.
- Occupational health and safety courses offered theoretically should be supported by practices and training and should be provided throughout working life. In this context, occupational health and safety courses should be offered in workshop environments and the training of experienced individuals should be emphasized.
- In OHS courses offered in the undergraduate department of universities, in order to better understand the risks that students will face during their professional lives, practical training should be started, and the course contents should be separated on the basis of branches in order to minimize the risks. From this point of view, it should be ensured that the issues are conveyed more comprehensively.
- In undergraduate education, toolbox training on OHS should be given regularly in laboratory courses and the efficiency of students should be made examinable.

**Author Contributions:** Conceptualization, M.Y. and I.O.; methodology, M.Y., I.K. and I.O.; validation, M.Y.; formal analysis, I.O., M.Y. and I.K.; investigation, I.O. and M.Y.; resources, M.Y. and I.O.; data curation, M.Y. and I.O.; writing—original draft preparation, I.O.; writing—review and editing, I.O. and M.Y.; visualization, I.O.; supervision, M.Y. and I.O.; project administration, M.Y. and I.K. All authors have read and agreed to the published version of the manuscript.

**Funding:** This research received no external funding.

**Institutional Review Board Statement:** The study was conducted in accordance with the Declaration of Helsinki and approved by the Ethics Committee of Institute of the Pure and Applied Sciences—Marmara University (E-44174047-302.10.01-394401; date of approval: 31 October 2022).

**Informed Consent Statement:** Participant consent was waived due to the nature of data collection, an anonymous online survey.

**Data Availability Statement:** The data presented in this study are available on request from the corresponding author.

**Acknowledgments:** Our thanks go to the Pure and Applied Sciences—Marmara University for advisory support in the planning phase of the study and for providing the contact details of the Marmara University. Furthermore, we would like to thank all participants in the study and the principals and teachers who forwarded the link to the online survey to the families.

**Conflicts of Interest:** The authors declare no conflict of interest.

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
