# Peer review of "Evaluation of Occupational Safety Courses Given in Undergraduate Departments in Turkey and Investigation of the Effect on Working Life"

_sustainability, doi:10.3390/su151612140_

Round 1
Reviewer 1 Report
Abstract
1. Revise the first sentence of the abstract as it seems syntactically awkward.
2. It would be more appropriate to use ‘the sample size of the study’ rather that the ‘universe of the study’
3. Does ‘easy sampling’ mean ‘convenient sampling?
4. The authors may wish to restructure the summary of the results to specifically explain what aspects of working life showed significant differences and what the significant correlations were. The authors are suggested to highlight the significant findings of the study rather than the general statistical trends and correlations.
5. The authors may wish to explain the practical implications of the study.
Introduction
1. ‘Occupational health safety’ should be ‘occupational health and safety’. Why is occupational health and safety a phenomenon? Wouldn’t it be a management practice?
2. The authors may explain why material and moral losses, deaths and diseases are important to social and public orders.
3. Page 1 – No references provided for the factors affecting number of occupational accidents and diseases.
4. Page 2 – Does ‘low-tech classrooms’ mean ‘low-tech class or category’?
5. Why low-tech class is associated with occupational incidences?
6. ‘Mandatory by international law’ should be ‘mandated by international law’
7. The authors claimed that occupational health and safety is important across all jobs but examine only the OHS courses offered to students of OHS major. What is the rationale for this?
8. The authors may wish to explain the connection between Higher Education Law and OHS Law.
9. The authors may wish to redefine the problem statement to clearly justify the need for this study, e.g., why is it important to evaluate the OHS courses of interest? What could be the problems related to those courses?
Occupational Health and Safety
1. Page 2 – revise ‘the worker and the worker are in harmony’.
2. Page 3 – ‘at least 3/1’ should be ‘at least 1/3’
3. This section contains repetitive information on the occurrences and impacts of occupational incidences, and the importance in addressing these incidences.
4. In-text citations are missing from a large part of the text.
5. This section contains general information about occupational health and safety which is rather overwhelming. The authors are suggested to streamline the contents to focus more on bringing out the research questions.
Training in Occupational Health and Safety
1. ‘Does natural right’ means ‘legal right’?
2. The authors brought up an important problem here concerning the failure of knowledge to translate to attitudinal and behavioral change but did not probe this further. This study, however, did not examine the discord between knowledge, attitude and behaviour further. Instead, it asserted that knowledge will increase the interest in OHS training which will lead to positive behavioral change. This is contradictory and lacks literature support.
3. The authors mentioned about habit but did not further define what it is, how it is developed and its impacts on OHS.
4. The authors may wish to support the claim that knowledge directly influences behaviour with strong theoretical framework as many studies and frameworks do not point to a direct relationship between knowledge and behavioral change. This is also the case in many studies related to climate change and environmental pollution.
5. Page 5 – ‘Does the 4-year OHS training mean 4-year OHS major’? Training is fundamentally different from majoring in OHS. Training can be attended by anyone who does not major in OHS.
6. Page 7 – Why is common sense an element to ensure the elimination of hazards? Common sense is subjective and may not work similarly for everyone.
7. Similarly, this section contains a large amount of general information about OHS training without relating it to what this study aims to examine.
8. The authors may consider performing literature review to illustrate the theoretical frameworks and highlight the research gaps instead.
Methods
1. The authors may wish to explain how the hypotheses were derived or illustrate the literature implying that the hypotheses are worth investigating. In other words, hypotheses are usually supported by certain theoretical frameworks.
2. It may not be necessary for the authors to explain what a quantitative study is. Rather, the authors may focus on the theoretical models underpinning the hypotheses.
3. Detail of the samples is lacking, e.g. undergraduate students of which courses? How many samples from each course? How many OHS subjects they took during their undergraduate years? What were the occupations of the graduate sample?
4. Information about ethical approval and research consent is missing.
5. Justification of the sample size is missing. How did the authors determine if the sample size was representative?
Conclusions
1. This section should be named ‘Results and Discussions’
2. Table 4 – While the analysis showed significance difference between the groups, what ere the effects that the authors were investigating? The relevance of the course? Improved safety performance? Perceived beneficial effects? Were the undergraduate able to respond to this question since they were not in the workforce yet?
3. Table 4 – Similarly, what were the aspects examined under occupational health and safety?
4. What were the implications in measuring demographic differences in the responses? The demographic differences do not really reflect the effectiveness of OHS courses.
5. How did the study show the differences in those who received OHS courses in various aspects of OHS such as being more aware of OHS issues in workplaces? How did OHS-learning duration affect OHS awareness, compliance, etc.? How did OHS major and non-OHS major differ in their responses to different OHS aspects. There are important dimensions related to effectiveness of a course that were apparently forewent.
Discussion
1. The inferences related to the relation between age and knowledge were not justified. OHS undergraduates might have more knowledge that non-OHS major who had more work experiences.
2. The authors may need to support their inferences with literature, and compare their findings with those published previously.
Conclusion and recommendations
1. How did the study prove that the undergraduate OHS education is not sufficient?
2. The conclusion and recommendations do not seem to tie with the findings of the study.
There are some grammatical and syntactical errors. The authors may wish to perform another round of thorough proofreading.
Author Response
Dear Reviewer,
We would like to thank you for the insightful comments and suggestions. We made all possible changes that were suggested and detailed the changes in the table below. Prior to response your comments we want to inform you that all the revisions and improvements are highlighted red the revised version of our manuscript. We sincerely appreciate your insightful comments on our paper.
We would like to thank you again for your valuable time and insight to strengthen our paper.
Yours truly,
Corresponding author on behalf of the authors
Dr. Ä°lyas Kartal
1.The authors claimed that occupational health and safety is important across all jobs but examine only the OHS courses offered to students of OHS majör. Whats is the rationale fort his?
2.The authors may wish to redefine the problem statement to clearly justify the need fort his study e.g., why is it important to evaluate the OHS courses of interest? What could be the problems related to those courses?
3.Does the 4-year OHS training mena 4-year OHS major”? Training is fundamentally different from majoring in OHS. Training can be attended by anyone who does not major in OHS. |
1. Since the research was limited to students who took occupational health and safety courses in undergraduate departments, not all occupational groups were included.
2. Since there is OHS training taken in undergraduate education instead of OHS courses in the research, no information was given about the problems related to the courses taken in OHS courses.
3. Engineering, science, architecture faculty graduates and technical teachers, occupational health and safety undergraduate or associate degree program graduates can become occupational safety experts by taking the OHS exam organized by OSYM. Engineering faculty graduates are entitled to take the exam after receiving 220 hours of training from OHS education institutions authorized by the Ministry of Labor and Social Security and 40 hours of internship. Candidates who reach 70 points in the exam are entitled to receive an occupational safety expert certificate. |
4.Detail of the samples is lacking e.g. undergraduate students of which courses? How many samples from each course? How many OHS subjects they took during their undergraduate years? What were the occupations of the graduate sample?
5.Why is occupational health and safety a phonomenon? Wouldn’t it be a management practice? |
4. Graduates include individuals who have graduated from engineering, architecture, science and occupational health and safety departments and those who do not have work experience before, such as internships, etc.
5. Occupational Health and safety can be considered as a management practice. Because employers implement policies and procedures to ensure the occupational health and safety of their employees and to protect them from accidents and occupational diseases. This situation may enable occupational health and safety to be seen as a management practice as well as a phenomenon. Employers and managers should establish occupational health and safety policies and procedures in the workplace, ensure employee training, conduct regular audits and assess risks. Occupational health and safety management consists of a set of methods, processes and strategies implemented by employers and managers to ensure the safety of employees and meet legal requirements. |

Reviewer 2 Report
This interesting study aimed to examine the evaluation of occupational safety courses given in undergraduate departments in Turkey and their effect on working life were examined. The authors investigated whether taking occupational safety courses as part of a bachelor's degree has an impact on the working life of persons who have finished their education and started working life.
The study's aim was to explore the effects of occupational health and safety courses on working life and occupational health and safety education according to employees’ occupational status, age, gender, and departments.
The present study tried to explore how the effect the occupational safety courses giving in undergraduate departments in Turkey on future working life of the employees.
The survey was attended by people with work experience as well as students, from several scientific fields, with and without experience of work accidents. Data were provided by both men and women. Thus, the sample can be examined in terms of established hypotheses.
The measurements and instruments used by the authors seem to be valid.
The results are processed in detail, with statistical confirmation or refutation of established hypotheses.
The discussion is a reasonable extent and includes the essential findings of the study. More literature can be added to the discussion, enriching the authors' arguments.
Limiting criteria are missing from the paper. Supplementing the limited criteria will increase the quality and the informative value of the study.
The paper I evaluate positively due to the confirmation of statistically significant differences between the effects of occupational health and safety courses on working life and occupational health and safety education according to the age of the participants. The conclusions show the inadequacy of the theoretical courses of safety and health protection at work provided within the bachelor programs at Turkish universities and the need to support them with practice and training in working life.
Author Response
Dear Reviewer,
We would like to thank you for the insightful comments and suggestions. We made all possible changes that were suggested and detailed the changes in the table below. Prior to response your comments we want to inform you that all the revisions and improvements are highlighted red the revised version of our manuscript. We sincerely appreciate your insightful comments on our paper.
We would like to thank you again for your valuable time and insight to strengthen our paper.
Yours truly,
Corresponding author on behalf of the authors
Dr. Ä°lyas Kartal

Reviewer 3 Report
Overall, this article is well-structured and understandable. Instead of listing, it would be great if the authors could classify the conclusions with comprehensive descriptions to make the findings more logical and precise.
Author Response

(The authors gave the same response as above.)

Reviewer 4 Report
Dear Authors,
After reviewing the article, I will proceed to list some comments so that you can improve its quality, since I consider it not publishable.
First, on the theoretical framework:
- I consider that the theoretical framework is poorly supported by concrete references. Only one reference is used to corroborate claims.
- I believe it is necessary to go a little deeper into the justification of the relevance of the study. Reinforce the idea of how the results would impact the population/students.
Second, on the materials and methods:
- I consider that the method and design is not properly explained. Very valuable information is missing about the type of study, the variables, the procedure, the sample (mean and standard deviation of age are not given), very basic information that is not necessary is provided.
- The general level of the method is very low.
- Many questionnaires are mentioned, but they are not identified, there is no way to corroborate the correspondence between the variables and the measures.
- It is not necessary to create a section on the validity/reliability of the instrument or on the normality of the responses; it is mentioned and that would be enough.
- The description of the data analysis is of poor quality; no specific technique, strategy or analysis is related to the established objectives or hypotheses.
- In general, the information is basic and not necessary due to a lack of relevant results.
In third place, in relation to results:
- Point 5 is not conclusions, it is results.
- The results are very basic since the analyses are basic as well. For example, it is not necessary to present descriptive results (PERCENTAGE AND FREQUENCY).
- In general, the data are presented in tables, but the most relevant data are not highlighted.
Fourth, regarding the discussion and conclusions
- I strongly recommend reinforcing the limitations of the study. They are rather superficial.
- I recommend revising the conclusions, it would be positive to include more integrated information and not so enumerated.
I hope you will be able to attend to these considerations, which are always aimed at improvement.
With my best wishes.
In relation to the English language, I consider that there are a few insignificant errors which, with a thorough revision, could be solved.
Author Response

(The authors gave the same response as above.)

Reviewer 5 Report
Evaluation of Occupational Safety Courses Given in Undergraduate Departments in Turkey and Investigation of the Effect on Working Life
Abstract
The abstract is unattractive, the abstract should follow the solid scientific work structure, I suggest that it be restructured in a way that includes the following clearly:
The contextualization of the study
· The main objective
· The justification
· The sample used
· The methods used
· The main findings and conclusions
· The novel contribution
Introduction
1.The introduction needs to be more clear and straight to the point by justifying soundly on the main objective; to examine the effect of occupational health and safety courses on working life, whether the contribution of occupational health and safety education and occupational health and safety education to awareness differs according to some variables and to determine whether there is a relationship between them.
2. What is the importance of occupational health and safety courses in the Undergraduate Departments in Turkey?
3. State the method at the end of the introduction, as well as the study's novel contributions.
4. The first three paragraphs need citation.
Literature Review
Literature section needs to be improved. The background section of the paper doesn’t demonstrate a clear relationship to the problem. Not enough literature has been provided. The sources cited are not enough. The research gaps in the previous studies related to this topic were not explained clearly. Please address these issues.
Methodology.
First the The hypothesis formation should not be under this section.
It is better to be at the end of Literature Review
Second.We need some answers:
How did the authors contact the respondents? Explain how the survey was distributed? What is the response rate? Does the authors' sample represent the population? Make sure whether the authors' sample can represent the population.
Conclusions
I think you meant Results.
This is not the right place to Conclusions
Discussions
1.The discussions are somewhat broad and general. I can’t see the clear explanation for this argument. It is complex to follow the line of arguing and to identify how each of the analyses and their results justifies the discussion.
2. I need to see the a clear discussion to the five Hypotheses.
3. Some aspects of the discussion are included in the results section.
Conclusion and Recommendations
The conclusions section should be restructured and shortened in a way that includes a clear theoretical implication, the practical implication of the research, future research, recommendations, and the limitation of the research.
Others:
The paper has some editing issues. It needs proofreading.
Good luck
The paper has some editing issues. It needs proofreading.
Author Response

(The authors gave the same response as above.)

Reviewer 6 Report
Article’s theme and contribution
Several topics have explored in the article.
Firstly, according to the one-way variance test conducted to compare the participants, statistically significant differences were found between the effects of occupational health and safety courses on working life and occupational health and safety education according to their occupational status, age, gender, and departments (p<0.05).
Secondly, statistically significant differences were found between the participants participating in the study and the occupational health and safety training according to the independent groups t-test, which was con- ducted to compare them according to their near misses and work experience during the training (p<0.05).
Then, according to the Pearson Correlation test conducted to examine the relationship between the effect of occupational health and safety courses on working life, occupational health and safety education and the contribution of occupational health and safety education to awareness, it was determined that there was a statistically significant relationship be- tween them (p<0.01).
Finally, no statistically significant differences were found according to the occupational accidents experienced by the employees (p>0.05).
Some comments/suggestions:
l The Study Hypotheses 4 and 5 are inconsistent with the following research design, which should be a misnomer, resulting in an error in semantic expression. There should be modified to statistically significant differences in the impact of occupational health and safety courses on working life, occupational health and safety education, and the contribution of occupational health and safety education to awareness in terms of department and work experience.
l The section of Discussion is slightly thin, especially regarding how gender affects occupational health and safety education, which can be further discussed in terms of women's psychological mechanisms.
The main confusion that exists:
l Is there a crossover between OHS courses and OHS education? How to distinguish the relationship between the two?
The structure of some paragraphs of the article:
l In the section 2 occupational health and safety, the second paragraph is a bit redundant and can be considered adjusting the location or deleting it outright.
l The abbreviation for what OHS means should be marked where it is first mentioned in the text.
l The relationship between occupational health and safety training and occupational health and safety education should be explained in the second part.
Please ask the author to find a professional or native speaker to polish the language。
Author Response
Dear Reviewers,
We would like to thank you for the insightful comments and suggestions. We made all possible changes that were suggested and detailed the changes in the table below. Prior to response your comments we want to inform you that all the revisions and improvements are highlighted red the revised version of our manuscript. We sincerely appreciate your insightful comments on our paper.
We would like to thank you again for your valuable time and insight to strengthen our paper.
Yours truly,
Corresponding author on behalf of the authors

Round 2
Reviewer 1 Report
While the paper has improved substantially from its previous version, there are still concerns that the authors may need to address as below:
1. Please check the punctuation to remove obvious errors, e.g. ‘compensation to be paid by the workplace..’, ‘ac-curate’, etc.
2. Please revise the sentence starting with ‘because low technologies cause an increase…’ as it is syntactically incorrect.
3. Please check the sentence ‘it is also mandated the risk…’
4. The authors may explain why convenience sampling is based on items which are readily accessible, fast and simple and why convenience sampling is the best sampling strategy for this study.
5. Please check the in-text citations again for the Section 2 and other sections as they seem to appear only at the end of each paragraph.
6. What are the ‘sub-dimensions of the awareness scale’?
7. The authors may elaborate further on the study of Korkutan (2018), particularly on how it is related to or inspired the current study.
8. The authors may consider organizing the newly added literature review to improve its coherence and highlight its relevance to this study.
9. Revise the paragraph starting with ‘When the curriculum of Yeni Yuzyil University is examined…’ as it seems grammatically flawed. Please also explain why all the subjects of the curriculum are listed.
10. Again, there is a lack of description of the theoretical framework used to draw the hypotheses. For instance, what is the theoretical framework that support the contribution of training to awareness as well as the demographic factors such as age, gender, department at play.
11. The authors may revise the sentence starting with ‘Although simplicity and materiality are important in research….’ to improve its clarity.
12. The authors may need to further justify the adequacy of sampling size based on statistical method since a bigger sample size should be adopted when the population becomes bigger.
13. The authors may present the tests of normality to justify the use of parametric analyses.
14. The authors may explain if opinions on OHS which correlated with age in the study of Sasar (2022) are the same or related to awareness investigated in this study.
15. The authors brought up a contrasting finding on the lack of differences of OHS views/awareness between department. The authors may explain why this finding is important in supporting the findings of this study.
16. There are still multiple grammatical, punctuation and syntactical errors, which the authors may need to correct before publication of the article. The authors may also consider improving the coherence of contents so it is easier for the readers to follow the idea.
The paper needs to be proofread thoroughly as there are still obvious language errors that require fixing.
Author Response
While the paper has improved substantially from its previous version, there are still concerns that the authors may need to address as below:
- Please check the punctuation to remove obvious errors, e.g. ‘compensation to be paid by the workplace..’, ‘ac-curate’, etc. ---------The edit has been made.
- Please revise the sentence starting with ‘because low technologies cause an increase…’ as it is syntactically incorrect. --------While low-tech causes increased workplace accidents, high-tech categories aim to minimize job accidents. (Page 1)
- Please check the sentence ‘it is also mandated the risk…’ ----------In addition, risks and hazards in the workplace necessitate occupational health and safety [3]. (Page 2)
- The authors may explain why convenience sampling is based on items which are readily accessible, fast and simple and why convenience sampling is the best sampling strategy for this study.---------------- It is based on ready-made, fast and easy-to-access items since random selection is made from a crowded universe and the most easily accessible participants constitute the sample. Since it was determined in our study that individuals with occupational health and safety training formed a crowded universe, making a random selection among them shows that convenient sampling is the best sampling strategy for this study. (Page 9)
- Please check the in-text citations again for the Section 2 and other sections as they seem to appear only at the end of each paragraph.------- In-text citations have been checked.
- What are the ‘sub-dimensions of the awareness scale’?-------- These questionnaires used do not consist of sub-dimensions, but consist of a single dimension.
- The authors may elaborate further on the study of Korkutan (2018), particularly on how it is related to or inspired the current study.------ It was removed from the literature after control.
Yenisarı et al. (2019), a descriptive field study was conducted to determine the awareness levels of employees on OHS training. In the study, literature review and survey methods, which are quantitative research techniques, were used. The survey was conducted by face-to-face interviews with academic and administrative staff working in different units of Çanakkale Onsekiz Mart University. The answers given by the participants to the questionnaires were subjected to confidence analysis using the Cronbach Alpha test. As a result of the research, approximately 12.8% of the par-ticipants stated that the causes of occupational accidents were due to lack of education [14]. (Page 4)
- The authors may consider organizing the newly added literature review to improve its coherence and highlight its relevance to this study.------- When the studies in the literature are examined, it is seen that there are studies examining Occupational Health and Safety education within the scope of different variables. While some of these studies show similarities with the study we discussed, some do not. It is important to examine the studies in the literature and to make the missing points the subject of new studies. (Page 3)
- Revise the paragraph starting with ‘When the curriculum of Yeni Yuzyil University is examined…’ as it seems grammatically flawed. Please also explain why all the subjects of the curriculum are listed.--------- In the previous revision, only some of the course contents were listed to provide information about the curriculum at the request of the referee number X.
- Again, there is a lack of description of the theoretical framework used to draw the hypotheses. For instance, what is the theoretical framework that support the contribution of training to awareness as well as the demographic factors such as age, gender, department at play.---------- Education is not a variable that is examined, since studies are conducted with individuals who receive occupational health and safety training at the undergraduate level.
- The authors may revise the sentence starting with ‘Although simplicity and materiality are important in research….’ to improve its clarity.------------------- In the research, it is taken into account where the most information can be obtained from the limited number of cases to be sampled, and the sample is selected accordingly [41]. (Page 9)
- The authors may need to further justify the adequacy of sampling size based on statistical method since a bigger sample size should be adopted when the population becomes bigger.------------ In addition, it was determined that the sample of the study was sufficient to represent the population with the gpower analysis performed with 95% confidence and 5% margin of error. (Page 9)
- The authors may present the tests of normality to justify the use of parametric analyses.--------- The kurtosis and skewness values obtained from the scale scores between +3 and -3 are considered sufficient for normal distribution. In addition, histograms, graphs and coef-ficients of variation for the relevant scale and its dimensions were also examined and found to be suitable for normal distribution. In this context, parametric methods were used in the analyses. ANOVA test was used in the analyses according to age and de-partment variables, independent samples t test was used in the analyses according to gender and work experience status, and Pearson Correlation test was used in their re-lations with each other. (Page 10)
- The authors may explain if opinions on OHS which correlated with age in the study of Sasar (2022) are the same or related to awareness investigated in this study. ---------- Similar to our study, ÅžaÅŸar (2022), who examined the effect of occupational health and safety on working life according to age, found a positive and significant relationship between the age of the participants and their views on occupational health and safety [44]. (Page 14)
- The authors brought up a contrasting finding on the lack of differences of OHS views/awareness between department. The authors may explain why this finding is important in supporting the findings of this study. ------------- This contradictory situation that arises according to the departments is due to the occupational health and safety department. As in our study, there is a significant difference in the sample, while there are those directly from the occupational health and safety department. (Page 15)
- There are still multiple grammatical, punctuation and syntactical errors, which the authors may need to correct before publication of the article. The authors may also consider improving the coherence of contents so it is easier for the readers to follow the idea.-----The article has been grammatically revised.

Reviewer 4 Report
Dear authors,
Thank you for your response. Thank you very much also for marking the changes made in another colour. This has made the revision process much quicker.
In relation to your responses and changes, I consider that the article has improved substantially. It is potentially publishable in the present form.
Congratulations for the great work done.
Best regards
As I said initially, the quality of the English is tight and expressions have been revised throughout the text, which has improved its comprehensibility.
Author Response
Dear Reviewer,
I am writing to express my sincere gratitude for your valuable feedback and insightful suggestions on my recent manuscript entitled 'Evaluation of Occupational Safety Courses Taught in Undergraduate Departments in Turkey and Investigation of Its Effects on Working Life', which was submitted to Sustainability. Your expertise and careful evaluation have significantly contributed to the improvement of my research.
Your comprehensive and constructive comments have been immensely helpful in enhancing the quality and clarity of my work. Your attention to detail and your ability to identify areas that required further development have undoubtedly strengthened the overall impact of the paper.
Furthermore, I would like to acknowledge your promptness in completing the review process.The academic community greatly benefits from the rigorous review process facilitated by esteemed reviewers like yourself.
Thank you once again for your invaluable contribution.
Yours truly

Reviewer 5 Report
Evaluation of Occupational Safety Courses Given in Undergraduate Departments in Turkey and Investigation of the Effect on Working Life
Abstract
Is fine followed the solid scientific work structure, but The abstract is long, I suggest that it be shortened to be more clear and attractive.
Introduction and Literature Review
Hypothesis formation should be under this section. It is better to be at the end of Literature Review
Methodology.
Hypothesis formation should not be under this section
The conclusions
Fine, but it is too long, I suggest that it be shortened to be more clear and attractive.
Good luck
Minor editing of English language required
Author Response
Comments and Suggestions for Authors
Evaluation of Occupational Safety Courses Given in Undergraduate Departments in Turkey and Investigation of the Effect on Working Life
Abstract
Is fine followed the solid scientific work structure, but The abstract is long, I suggest that it be shortened to be more clear and attractive.------------------ In this study, using the general survey model, which is one of the quantitative research methods, the evaluation of the occupational safety courses given in undergraduate departments in Turkey and their effects on working life were examined with 440 people who received their undergraduate education and started their working life. Appropriate sampling method was preferred in the sample selection of the study. According to the research, the Effect of Occupational Health and Safety Courses on Working Life, the Contribution of Occupational Health and Safety Education and Occupational Health and Safety Education's Contribution to Awareness show significant differ-ences according to age, gender, department and work experience. When the relationship between them is examined, these relationships are positive and moderate. The results and survey average scores show that the theoretical courses on occupational health and safety in undergraduate pro-grams of universities in Turkey are insufficient and should be supported by practice and training in working life.
Introduction and Literature Review
Hypothesis formation should be under this section. It is better to be at the end of Literature Review ---------------- The hypotheses were added to the end of the related studies.
Methodology.
Hypothesis formation should not be under this section ------------------------- The hypotheses were removed from the method section.
The conclusions
Fine, but it is too long, I suggest that it be shortened to be more clear and attractive.--------------- In this study, which aims to evaluate the occupational safety courses given in undergraduate departments in Turkey and to examine their impact on working life, it was investigated whether taking occupational safety courses within the scope of undergraduate education has an impact on the working lives of people who finish their education and start working life. In the study, the effects of occupational health and safety courses on working life and occupational health and safety education were examined according to the occupational status, age, gender and department of the employees. Students from various disciplines with and without work accident experience and people with work experience participated in the survey. Data were collected by both men and women. According to the research, the Effect of Occupational Health and Safety Courses on Working Life, the Contribution of Occupational Health and Safety Education and Occupational Health and Safety Education's Contribution to Awareness show significant differences according to age, gen-der, department and work experience. When the relationship between them is examined, these relationships are positive and moderate. The results obtained and the average scores of the questionnaire show that the theoretical courses on the protection of health and safety in the workplace in undergraduate programs in universities in Turkey are insufficient and should be supported by practice and training in working life. The fact that the research was conducted only in Turkey and that the data was collected online poses a limitation in terms of generalizability.
